# qPCR in a suitcase for rapid *Plasmodium falciparum* and *Plasmodium vivax* surveillance in Ethiopia

Lise Carlier[1,2�], Sarah Cate Baker[1,3�], Tiffany Huwe[4], Delenasaw Yewhalaw[5], Werissaw Haileselassie[6], Cristian Koepfli[4]*

**1** Trinity College Dublin, Dublin, Ireland, **2** Foundation for Innovative New Diagnostics, Geneva, Switzerland, **3** Oregon Health & Science University, Portland, Oregon, United States of America, **4** Department of Biological Sciences & Eck Institute for Global Health, University of Notre Dame, Notre Dame, Indiana, United States of America, **5** Tropical and Infectious Disease Research Center, Jimma University, Jimma, Ethiopia, **6** School of Public Health, Addis Ababa University, Addis Ababa, Ethiopia

☉ These authors contributed equally to this work.
* ckoepfli@nd.edu

**Data Availability Statement:** All data are included in the paper and the Supporting Information file titled S1 Database.

## Abstract

Many *Plasmodium* spp. infections, both in clinical and asymptomatic patients, are below the limit of detection of light microscopy or rapid diagnostic test (RDT). Molecular diagnosis by qPCR can be valuable for surveillance, but is often hampered by absence of laboratory capacity in endemic countries. To overcome this limitation, we optimized and tested a mobile qPCR laboratory for molecular diagnosis in Ziway, Ethiopia, where transmission intensity is low. Protocols were optimized to achieve high throughput and minimize costs and weight for easy transport. 899 samples from febrile patients and 1021 samples from asymptomatic individuals were screened by local microscopy, RDT, and qPCR within a period of six weeks. 34/52 clinical *Plasmodium falciparum* infections were missed by microscopy and RDT. Only 4 asymptomatic infections were detected. No *hrp2* deletions were observed among 25 samples typed, but 19/24 samples carried *hrp3* deletions. The majority (25/41) of *Plasmodium vivax* infections (1371 samples screened) were found among asymptomatic individuals. All asymptomatic *P. vivax* infections were negative by microscopy and RDT. In conclusion, the mobile laboratory described here can identify hidden parasite reservoirs within a short period of time, and thus inform malaria control activities.

## Introduction

Malaria remains a major public health threat in many countries in the tropics and subtropics. After a decade of progress with a pronounced reduction of the number of clinical cases and deaths, progress has stalled in recent years. In 2020, over 240 million cases and 600,000 deaths were recorded [1].

Accurate and fast diagnosis and treatment are key aspects of malaria control. In most malaria-endemic countries, diagnosis by light microscopy is routinely conducted at health centers and hospitals. The sensitivity and specificity of local microscopy depends greatly on the training of

**Funding:** This work was supported by NIH R21AI137891 (CK). The funders had no role in study design, data collection and analysis, decision to publish, or preparation of the manuscript.

local microscopists [2], and field microscopy can be substantially less sensitive than expert micros-copy [3]. As an alternative, rapid diagnostic tests (RDTs) have become increasingly common. RDTs are lateral flow devices that detect parasite-specific proteins through immunohistochemis-try. RDTs require less training, and results are obtained within 10 minutes. They are thus used by small health posts with no microscopy infrastructure and by health workers conducting house-hold visits and diagnosis, e.g. in the frame of reactive case detection activities [4]. Sensitivity of RDTs can be impaired by incorrect storage and handling, wrong interpretation of results, or dele-tion of the gene coding for Histidine-Rich Protein 2 (HRP2), which is detected by most RDTs for *P. falciparum* [5]. False-positive results can be caused by non-malarial infections [6].

Light microscopy and RDT have a limit of detection of approximately 50–100 parasites per uL of blood [7]. A large number of clinical infections remain below this density [8,9]. Further, in all transmission settings, a proportion of infections remain asymptomatic, and many of them are subpatent [10,11]. Asymptomatic infections and low-density clinical infections escaping routine diagnosis conducted at health centers among febrile patients sustain trans-mission and present a major challenge to control [12–15].

Molecular diagnosis by PCR or other nucleic acid amplification tests are required to assess the quality of local diagnosis, to determine the true number of infections among febrile patients, and to understand population parasite prevalence in asymptomatic individuals. Rapid, sensitive screening might also be required to coordinate the response to outbreaks, for example to decide where intensified vector control is warranted because of a large asymptomatic reservoir. Molecu-lar surveillance is often complicated by the absence of laboratory infrastructure and lack of skilled personnel in malaria endemic sites. Shipment of samples to reference laboratories can be compli-cated and time consuming. Molecular screening is thus seldom applied to select control strategies tailored to local conditions, or in response to outbreaks.

In order to speed up time to result and enable in-country scientists and control programs to process samples, efforts are increasingly being made to bring laboratory capacity to field sites [16–18]. Numerous devices and protocols for molecular screening for pathogens are being developed and trialed. Often, these assays rely on custom-built devices [19]. Throughput of commercially available platforms is often low [20–24]. In addition, the need for high-throughput, mobile DNA extraction platforms is not addressed.

For this study, a mobile qPCR lab was trialed for malaria surveillance in a low transmission site in Ethiopia. All equipment and consumables needed are commercially available and fit in suitcases for transport on airplanes. Up to two 96-well plates can be processed in a day, at a cost of approximately USD 2.5 per sample for DNA extraction and *P. falciparum* and *P. vivax* qPCR. Within a brief period of 2 months, nearly 2000 samples from febrile cases and asymp-tomatic individuals were screened using highly sensitive qPCR.

## Methods

### Ethical approval

Informed written consent was obtained prior to sample collection from each study participant or, in the case of minors, from their parent or legal guardian. The study protocol was approved by the University of Notre Dame IRB (#19-03-5201), Trinity College Dublin, Addis Ababa University, and the National Research Ethics Review Committee at Ministry of Science and Higher education (MoSHE).

### Study site

In Ethiopia, *P. falciparum* and *P. vivax* are endemic. Malaria transmission ranges from very high in the tropical lowlands along the borders with Sudan and South Sudan to low and

sporadic in the highlands [25]. In 2019, over 900,000 confirmed cases were reported. This represents a pronounced reduction compared to 2013, where the number of cases peaked at 2.6 million, but only a moderate reduction compared to 2010, with 1.2 million confirmed cases [26]. Diagnosis is provided at over 20,000 health centers across the country. Larger health centers perform diagnosis by microscopy, while RDTs are used by smaller health posts. In addition, over 70,000 health extension workers visit households and provide basic medical services, including malaria diagnosis by RDT [27].

This study was conducted in Ziway, Oromia region. Transmission intensity of *P. falciparum* and *P. vivax* is low. Samples for the current study were collected in the low transmission season in June and July 2019. Clinical samples were collected from individuals presenting with febrile illness to Batu and Dembel Health Centers. Cross-sectional surveys were conducted in 3 rural kebele (the lowest administrative units in Ethiopia), Bochessa, Dodicha, and Golba, which are under Adami Tulu Jiddo Kombolcha district administration.

### Sample and data collection

For the clinical samples, patients with suspected malaria infection were invited to join the study and provide an additional blood sample for diagnosis. A brief questionnaire was completed including age, sex, and kebele of residence of the patient. For community samples, a convenience sampling strategy was applied. The study team visited the villages, approached households, and asked all household members who were present to provide a sample. 100–200 μL blood were collected by finger prick into EDTA tubes. Blood samples were stored on ice packs in Styrofoam boxes until bringing them to the lab each evening, where they were stored at -20˚C. RDT positive individuals among the community samples were referred to their health center for further diagnosis and treatment.

### Diagnosis by microscopy, RDT, and qPCR

Samples were collected by finger prick into EDTA tubes. All samples were screened by RDT (AccessBio CareStart Pf(HRP2)/Pv(LDH) combo) upon collection, and by local microscopy. For microscopy, WHO protocols were followed. 100 fields were assessed before declaring a sample negative.

DNA extraction was done using the Macherey-Nagel NucleoMag kit according to manufacturer's instructions, with the following modification (Box 1xy): As the kit is optimized for extraction from 200 uL blood, but DNA was extracted from only 100 uL of blood, the volume of all reagents was reduced by 50%. Thus, per kit 8x 96 samples could be extracted, further reducing the amount of materials required and cost per sample. As proposed by the manufacturer as option, after the ethanol wash-step, beads were air-dried for 15 minutes instead of using buffer MBL-4. Some of the volumes of buffers were slightly modified to be able to complete all steps with a 30–300 μL multichannel pipette (S1 Protocols). In a recent side-by-side comparison, the extraction kit used yielded significantly more DNA than a spin-column based kit [28].

qPCR was done in a total volume of 12 μL, including 4 μL DNA, corresponding to 4 μL blood. For *P. falciparum* qPCR the *varATS* assay was used. This assay targets a multicopy gene that is present in 10–20 copies per parasite [29]. Using the extraction method we chose for this study, the limit of detection of this assay is 0.3 parasites/μL blood [28]. *P. vivax* qPCR was done using the *cox1* assay. This assay targets a mitochondrial gene that is present in approximately 10 copies per parasite [30]. Detailed qPCR protocols are given in S1 Protocols. Due to a manufacturing problem with the *P. vivax* probe (low yield), only a random subset of samples was screened by qPCR (653/1021 asymptomatic and 718/899 clinical samples). Infection

prevalence and test positivity rate was compared among three age groups of 0<5 years, 5<15 years, and ≥15 years, between males and females, and between kebele (community sampling only) using Pearson's Chi-square test.

HRP2-based RDTs can also detect the HRP3 protein, though sensitivity is lower [31]. Deletions of the *hrp2* gene result in false-negative RDTs in low-medium density infections. Deletions of *hrp2* and *hrp3* result in negative HRP2-based RDTs irrespective of parasite density [32,33]. *P. falciparum* positive samples were typed for *hrp2/3* deletions by droplet digital PCR (ddPCR). In this assay, either *hrp2* or *hrp3* is multiplexed with a control gene, *serine-tRNA ligase*. Both targets are amplified with very high sensitivity and specificity, thus providing highly accurate data on deletion status [34]. For ddPCR, samples were shipped to the University of Notre Dame.

## Results

### Mobile laboratory

The mobile DNA extraction and qPCR systems were established in a makeshift laboratory on the compound of Addis Ababa University in Ziway. It consisted of a basic shed with two simple tables, and thus is representative for many locations with no laboratory infrastructure. All equipment and consumables required for this study are commercially available and given in Table 1. All protocols were optimized to achieve high throughput, i.e. work in 96-well format for extraction and 48-well format for the qPCR, while maintaining a low weight of the instruments required. Most importantly, the need for low weight instruments precluded the use of a centrifuge as used for common spin-column DNA extraction protocols. Instead, a protocol based on magnetic beads was used.

The main equipment required include the MIC qPCR system (including laptop computer), one plate shaker (for binding of DNA to the beads, wash steps, and DNA elution), one magnetic block (to bind the beads and remove the supernatant), two 12-channel pipettes (1–10 μL, 30–300 μL), and one set of single channel pipettes (1–20 μL, 20–200 μL, 100–1000 μL) (Fig 1). The total weight of all equipment was 7.2 kg, and it fits into one carry-on bag for air transport. The cost for all instruments totals approximately USD16,900 (not including the laptop computer). Costs are as follows: qPCR instrument: USD13,000, plate shaker: USD300, magnetic block: USD700, 2 x multichannel pipette: $1000 per pipette, 3 x single channel pipette: USD300 per pipette. The weight of all consumables for 8 plates (8x96 samples) was approximately 9.3 kg and thus can be easily transported by air as check-in luggage (Table 1). The cost of consumables including extraction kit, pipettes tips and other plasticware, and qPCR reagents was approximately USD 2.5 per sample.

The following items were purchased from local pharmacies: 99% Ethanol, nitrile gloves, lancets for blood sample collection, and microscopy slides. Further, a -20˚C freezer was purchased locally for storage of reagents and samples. Plastic buckets were purchased as waste bins. All extractions were done at room temperature. No water bath or incubator was needed.

### *P. falciparum* screening

Among 899 samples collected from febrile patients presenting to clinics, 55 (5.8%) were positive by qPCR. By local microscopy, only 13/52 qPCR positive samples were correctly diagnosed. One of the samples positive by qPCR for *P. falciparum* was misdiagnosed by microscopy as *P. vivax*. RDT was moderately more sensitive, with 18/52 qPCR positive samples detected by RDT. A total of six samples were positive by microscopy and/or RDT, but not confirmed by qPCR (Fig 2). They remained negative when the qPCR was repeated. Mix up at

**Table 1. Equipment and consumables required for qPCR screening.** One extraction kit lasts for 8 x 96-well plates. The number and weight of all consumables and reagents reflects quantities required for 8 x 96 samples.

| | Brand/type used | | Weight in g | Comments |
|---|---|---|---|---|
| **Equipment** | | | | |
| qPCR instrument | Biomolecular Systems | | 3750 | Weight does not include laptop computer |
| Plate shaker | USA scientific | | 1900 | No longer available. Alternative: SCILOGEX MX-M |
| Magnetic block | Macherey Nagel NucleoSep | | 500 | |
| Multichannel pipette 1–10 µL | USA scientific | | 200 | |
| Multichannel pipette 30–300 µL | USA scientific | | 200 | |
| Set of single channel pipettes | USA scientific | | 300 | |
| Tube rack | USA scientific | | 270 | Catalogue no. 2396–5001 |
| **Total weight of equipment [g]** | | | **7120** | |
| **Consumables** | **Brand/type used** | **Number required per 8 plates** | **Weight in g** | Comments |
| EDTA tubes | | | 1350 | 8x100 tubes |
| DNA extraction kit | Macherey Nagel NucleoMag® Blood 200 µL | 1 | 1800 | Buffer MBL-4 removed |
| 30–300 µL filter tips (refill racks of 96 tips) | USA scientific | 40 | 3000 | A few racks to use with the refills were packed |
| 300 µL TipOne graduated tip, bulk | USA scientific | 1 | 370 | |
| Reservoir for reagents | ThermoFisher | 16 | 100 | One used for extraxtion, one used for qPCR |
| DNA storage plate | ThermoFisher AB 0765 | 8 | 670 | |
| Strip caps | ThermoFisher AB 0981 | 8 | 50 | |
| Falcon tube 15 mL | VWR | 8 | 80 | To prepare proteinase/MBL1 mixture |
| Falcon tube 50 mL | | 8 | 100 | To prepare 80% ethanol |
| 1 mL filter tips | VWR | 8 | 185 | |
| Eppendorf tube | USA scientific | 16 | 20 | To prepare qPCR mastermix |
| Primers and probes | Sigma/Thermo Scientific | | 50 | |
| PCR-grade H$_2$O | | | 100 | To prepare 80% ethanol, primer and probe working solutions, and qPCR mastermix |
| PerfeCTa® qPCR ToughMix®, Low ROX™, QuantaBio | VWR | 4 | 8 | 1.25 mL per tube, 6 µL per reaction. Weight does not include ice packs. |
| MIC qPCR tubes | Genesee Scientific | 32 | 360 | 2x 48-well racks per DNA plate, 2 qPCRs (Pf/Pv) per plate. Tubes are pre-packed in racks. |
| 10 uL filter tips | USA scientific | 18 | 1100 | 2x12 tips to load reaction mixes, plus 1 rack each to load Pf and Pv qPCR plate. |
| **Total weight of consumables [g]** | | | **9343** | |

the health center during sample and data collection might have occurred. Demographic and qPCR data are given in Table 2. No significant difference in qPCR positivity by age group or sex was observed.

*P. falciparum* prevalence among asymptomatic individuals was very low with only 4/1021 (0.4%) individuals positive by qPCR. None of them were positive by microscopy or RDT. One individual tested positive by RDT, but the infection was not confirmed by qPCR. The four qPCR positive individuals were 4, 14, 15, and 40 years old; two were male and two were female. Three of those positive were from Golba, and one from Dodicha (Table 2).

25 *P. falciparum* positive samples were successfully typed for *hrp2* deletion, and 24 samples for *hrp3* deletion. No *hrp2* deletions were observed, but 19/24 samples lacked the *hrp3* gene.

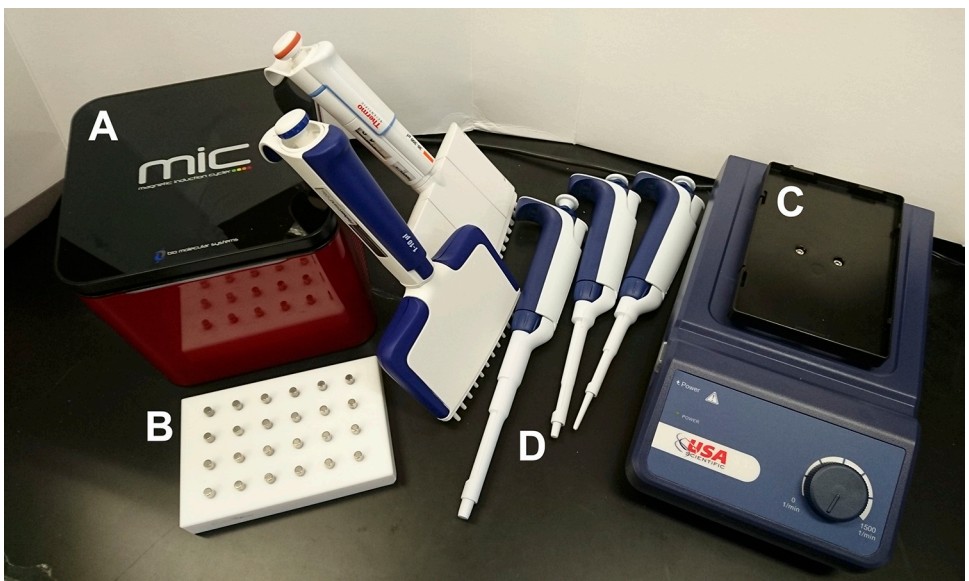

**Fig 1. Instruments required for mobile qPCR.** A) MIC 48-well qPCR instrument, B) magnetic block, C) plate shaker, D) pipettes.

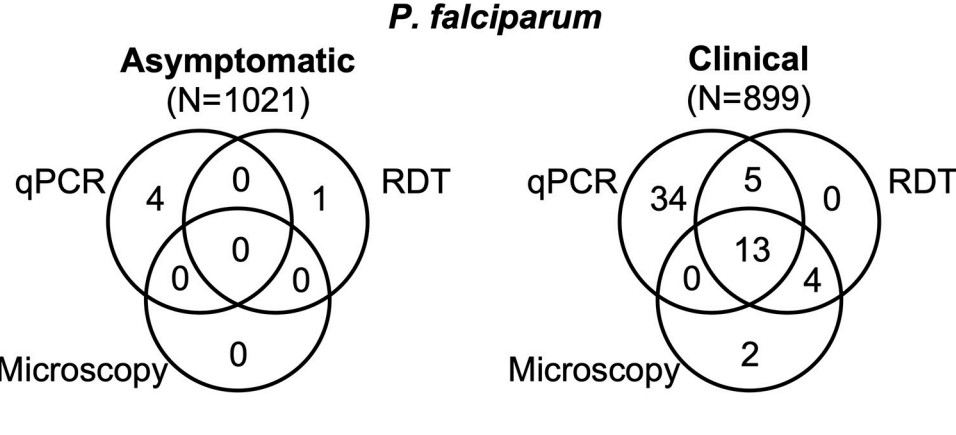

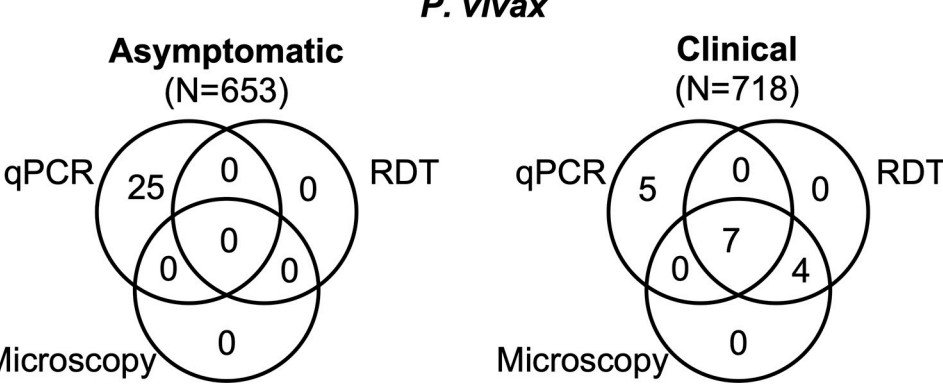

**Fig 2. Number of samples positive by qPCR, RDT, and microscopy among clinical and asymptomatic individuals.**

**Table 2. Demographic and qPCR data of study population.**

| Clinical patients | | | | | | |
|---|---|---|---|---|---|---|
| | | *P. falciparum* | | | *P. vivax* | |
| | n | qPCR positive | *P* | n | qPCR positive | *P* |
| **Age group** | | | | | | |
| <5 | 89 | 4 | | 73 | 1 | |
| 5≤15 | 197 | 15 | 0.435 | 163 | 1 | 0.443 |
| >15 | 613 | 33 | | 482 | 10 | |
| **Sex** | | | | | | |
| Male | 441 | 24 | | 354 | 5 | |
| Female | 453 | 28 | 0.637 | 347 | 7 | 0.544 |
| **Community sampling** | | | | | | |
| | | *P. falciparum* | | | *P. vivax* | |
| | n | qPCR positive | *P* | n | qPCR positive | *P* |
| **Age group** | | | | | | |
| <5 | 133 | 0 | | 88 | 4 | |
| 5≤15 | 413 | 2 | 0.732 | 276 | 9 | 0.8 |
| >15 | 475 | 2 | | 289 | 12 | |
| **Sex** | | | | | | |
| Male | 649 | 2 | | 422 | 16 | |
| Female | 372 | 2 | 0.572 | 231 | 9 | 0.947 |
| **Kebele** | | | | | | |
| Bochessa | 312 | 0 | | 196 | 1 | |
| Dodicha | 399 | 1 | 0.219 | 335 | 21 | 0.003 |
| Golba | 310 | 3 | | 122 | 3 | |

### *P. vivax* screening

Among 718 samples collected from febrile patients presenting to clinics and screened for *P. vivax* by qPCR, 12 (1.7%) were positive (Fig 2). Seven out of these twelve samples were also positive by microscopy and RDT. The twelve individuals that were positive by qPCR were 3 to 27 years old; with 10/12 being 15 years and older (Table 2). Four more samples were positive by microscopy and RDT, but not confirmed by qPCR. As for qPCR-negative/microscopy or RDT positive *P. falciparum* samples, they remained negative when the qPCR was repeated.

*P. vivax* prevalence among asymptomatic individuals was 3.8% (25/653). None of them was positive by microscopy or RDT (Fig 2). The age of positive individuals ranged from 1.5 to 60 years. No significant difference in prevalence rate among age groups was observed (Table 2). Prevalence differed significantly among kebele (*P* = 0.003). It was highest in Dodicha at 6.3% (21/335), and lower in Golba at 2.5% (3/122) and Bochessa at 0.5% (1/196) (Table 2).

## Discussion

Molecular screening for infections below the limit of detection of microscopy or RDT is a key component of molecular malaria surveillance, and often the first step for subsequent studies, such as parasite genotyping to quantify drug resistance or establish transmission networks. Lack of adequate laboratory infrastructure is a problem in many endemic countries. In this study, using a mobile qPCR setup, high quality data on *P. falciparum* and *P. vivax* infection status with a limit of detection of <1 parasite µL blood was obtained from almost 2000 samples within a period of a few weeks. A short turnaround time of a few weeks is required in order to

integrate molecular surveillance into control activities, for example to determine the extent of the asymptomatic reservoir during an outbreak [35].

The protocol tested is fully based on commercially available instruments and reagents, and offers high throughput with DNA extraction done in 96-well format, and qPCR run in 48-well format. A pair of trained laboratory technicians can process two 96-well plates within a day, and thus screen approximately 180 samples plus controls. The two-month study period included laboratory setup (e.g. cleaning, and procurement of freezer), visits to health centers and communities before sampling began, and sample collection. The limiting factor in the present study was the number of individuals that could be sampled per day. The cost of all consumables for DNA extraction, and separate *P. falciparum* and *P. vivax* qPCR, is approximately 2.5 USD per sample. The protocol used is suited for extraction of any DNA or RNA from blood and thus can be applied for molecular diagnosis of any blood-borne pathogen. The protocol requires multiple pipetting steps and thus molecular laboratory skills are needed. Likewise, knowledge is required to interpret qPCR data. Training of malaria control program personnel will be crucial in order to integrate qPCR data into routine surveillance activities. Based on training experiences in multiple malaria endemic countries, individuals with no prior laboratory experience can learn the skills to conduct DNA extractions and qPCR within two weeks (C. Koepfli, unpublished).

The makeshift laboratory presented multiple challenges. No running water was available, and power supply was unreliable with regular power cuts lasting several hours. As a result, the PCR was often run in the hotel, which had a back-up generator. For future surveillance by control programs, use of a generator to power the mobile lab is recommended. Of note, the DNA extraction can be done without the plate shaker, thus not requiring any power. Mixing steps can be done by pipetting. This protocol requires substantially more tips. The main risk of high-throughput manual DNA extraction in 96-well format is cross-contamination. This risk does not differ in a field laboratory compared to the same protocol being used in a reference laboratory, and can be minimized by proper training of personnel. The challenges of the mobile lab were offset by the rapid availability of data. This was highlighted by the extended period of time required to obtain permit to ship samples to the US for *hrp2* and *hrp3* deletion typing.

This study revealed crucial reservoirs for transmission not identified by current control. Two thirds (34/52) of *P. falciparum* infections detected by qPCR in febrile patients were missed by microscopy and RDT. These untreated infections likely contribute to transmission for an extended period of time [36]. More sensitive diagnostic tools at health centers would be expected to reduce transmission. In contrast, very few infections were detected among asymptomatic individuals. A contrasting pattern was observed for *P. vivax*. Most infections were detected among asymptomatic individuals, and fewer among febrile patients. Asymptomatic *P. vivax* infections clustered mostly on one kebele. Possibly, many of the asymptomatic infections were relapses. Yet, while asymptomatic, they can still contribute to transmission [37]. This study corroborated high rates of subpatent *P. falciparum* and *P. vivax* infections in Ethiopia [38,39]. Of note, transmission intensity in the Ziway region has declined drastically since 2005/2006, when a prevalence by microscopy of 16–19% was recorded [40].

In conclusion, this proof of concept study showed that actionable data on subpatent *P. falciparum* and *P. vivax* infections can be obtained in a short period of time using a mobile qPCR lab. Molecular screening has identified a gap in the sensitivity for diagnosis of clinical *P. falciparum* cases and a substantial asymptomatic *P. vivax* reservoir, which was mostly concentrated in one village. *P. falciparum* control should focus on more sensitive diagnosis in health centers, e.g. though the introduction of novel, ultra-sensitive rapid diagnostic tests [41]. *P. vivax*

control also needs to focus on prevention of onward transmission from the asymptomatic reservoir.

## Supporting information

**S1 Protocols. Laboratory protocols.**
(DOCX)

**S1 Database. Database.**
(XLSX)

**S1 Questionnaire. PLOS questionnaire on inclusivity in global research.**
(DOCX)

## Acknowledgments

We thank the study participants, kebele leaders, staff of AAU at Ziway satellite campus, staff at Batu and Dembel Health Centers, and the field collection teams.

## Author Contributions

**Conceptualization:** Lise Carlier, Sarah Cate Baker, Werissaw Haileselassie, Cristian Koepfli.

**Data curation:** Cristian Koepfli.

**Formal analysis:** Cristian Koepfli.

**Funding acquisition:** Cristian Koepfli.

**Investigation:** Lise Carlier, Sarah Cate Baker, Tiffany Huwe, Werissaw Haileselassie, Cristian Koepfli.

**Methodology:** Cristian Koepfli.

**Project administration:** Cristian Koepfli.

**Supervision:** Delenasaw Yewhalaw, Werissaw Haileselassie, Cristian Koepfli.

**Visualization:** Cristian Koepfli.

**Writing – original draft:** Cristian Koepfli.

**Writing – review & editing:** Lise Carlier, Sarah Cate Baker, Tiffany Huwe, Werissaw Haileselassie, Cristian Koepfli.

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
