## [Decision Letter · Decision Letter 0]

30 May 2022

PGPH-D-22-00611

qPCR in a suitcase for rapid Plasmodium falciparum and Plasmodium vivax surveillance in Ethiopia

Dear Dr. Koepfli,

Thank you for submitting your manuscript to PLOS Global Public Health. After careful consideration, we feel that it has merit but does not fully meet PLOS Global Public Health’s publication criteria as it currently stands. Therefore, we invite you to submit a revised version of the manuscript that addresses the points raised during the review process.

Please submit your revised manuscript by . If you will need more time than this to complete your revisions, please reply to this message or contact the journal office at globalpubhealth@plos.org. Please include the following items when submitting your revised manuscript:

We look forward to receiving your revised manuscript.

Kind regards,

Abhinav Sinha, M.D.

Academic Editor

Journal Requirements:

2. Please ensure that details in the Funding Information and Financial Disclosure Statement are matched.

3. Please provide separate figure files in .tif or .eps format.

4. We do not publish any copyright or trademark symbols that usually accompany proprietary names, eg (R), (C), or TM  (e.g. next to drug or reagent names). Please remove all instances of trademark/copyright symbols throughout the text, including ® on Table 1.

5. All figures and supporting information files will be published under the Creative Commons Attribution License (creativecommons.org/licenses/by/4.0/). Authors retain ownership of the copyright for their article and are responsible for third-party content used in the article. 

Fig 1: Please confirm (a) that you are the photographer; or (b) provide written permission from the photographer to publish the photo(s) under our CC-BY 4.0 license.

Please upload any written confirmation as an 'Other' file type. It must clarify that the copyright holder understands and agrees to the terms of the CC BY 4.0 license; general permission forms that do not specify permission to publish under the CC BY 4.0 will not be accepted. Note that uploading an email confirmation is acceptable.

Additional Editor Comments (if provided):

Reviewers' comments:

Reviewer's Responses to Questions

**Comments to the Author**

1. Does this manuscript meet PLOS Global Public Health’s publication criteria? Is the manuscript technically sound, and do the data support the conclusions? The manuscript must describe methodologically and ethically rigorous research with conclusions that are appropriately drawn based on the data presented.

Reviewer #1: Partly

Reviewer #2: Yes

2. Has the statistical analysis been performed appropriately and rigorously?

Reviewer #1: Yes

Reviewer #2: I don't know

3. Have the authors made all data underlying the findings in their manuscript fully available (please refer to the Data Availability Statement at the start of the manuscript PDF file)?

Reviewer #1: Yes

Reviewer #2: Yes

4. Is the manuscript presented in an intelligible fashion and written in standard English?

Reviewer #1: Yes

Reviewer #2: Yes

5. Review Comments to the Author

Reviewer #1: The author here tried to demonstrate the adaptability of a portable suitcase-based laboratory facility, which is able to detect the occurrence of genetic material of Plasmodium falciparum and P. vivax at a makeshift laboratory in Ethiopia. This portable facility is projected as a rapid tool for sensitive surveillance of malaria parasite infection and may be adapted in malaria control activities. However, the author, in addition to demonstration of the portable lab facility and its comparability with that of traditional diagnosis methods like microscopy and RDT; they could have validated the sensitivity and specificity of qPCR assay with some controls at the field site. Below are few concerns which should be clarified before the article get published.

Minor Revision

Line 150 typing error in 12-channel pipettes volume mentioned 1-20ul instead of 1-10ul as seen in fig.1

Line 150 typing error in volume mentioned 20-300ul instead of 30-300ul as seen in fig.1

Line 208 may clarify the “high quality data”

Line 209 may change from “a period of few week” to precise 2-months which is mentioned in intro section earlier.

Table.1 should mention the waste-bin and microtube racks.

Table.1 should mention the steps at which filter tips were used.

Methods should mention the PCR condition so that the time assessment of the qPCR will be clear.

Major Revision

1. Results described for P. falciparum screening was not matched with that of Venn diagram in fig.2 qPCR positives mention 51 instead of 52 as per fig.2; 13 positives from microscopy instead of 19 as per fig.2; 18 positives by RDT instead of 21 as per fig.2. qPCR positives in asymptomatic cases 4/1029 instead of 4/1021 as per fig.2.

2. Table.2 seems not that important and can be in supplementary files.

3. HRP-2/3 typing was not mentioned in the methods and seems to be performed at the central lab. The whole method of this typing should be mentioned and the results should be described in details.

4. Why 4 sample with positive P. vivax results for microscopy and RDT were repeated for the qPCR and the same repetition is not done for 6 sample with positive P. falciparum results for microscopy and RDT.

5. What is the “short turnaround time” in line 209, author should describe it.

6. Contradictory statement at different lines; line 213 mentioned need of pair of laboratory technicians enough to process for 2-plates/day, however line 217 mentioned molecular laboratory skills are needed to perform all these.

7. The sensitivity and specificity are not experimented here in this research work. Even if the published primer and probes is adapted here, there should be an explanation of any effect on the whole assay due to reducing DNA extraction from 200ul to 100ul initial source of DNA. May be all of the positives and few negatives from the field sites should be rechecked at the central lab using the isolated DNA from field sites for showing accuracy and reproducibility of this qPCR assay.

8. Why parasitaemia is not noted in this study?, as that may have helped in concluding the detection limit.

9. Why the whole assay of qPCR is singleplex, which seems to consume more time?

10. Few discussions are needed for the insight to the successful demonstration of suitcase lab;

a) What are the chances of false positives at field site due to cross contamination? What is the measure to avoid it in this kind of setting like qPCR suitcase-lab?

b) Who performed all qPCR for n=1920 with what skill sets? and how could be those skills be adapted by the technicians should be discussed.

c) The n=1920 samples were successfully performed in 2 months. Here, through is presented as 2*96=192/day and according to that n=1920 should be completed in 10 days. So, it should be clearly mentioned the throughput because current throughput seems around 192/day DNA isolation and 48/day qPCR to complete n=1920 sample in 2-months.

d) The limitation may be mentioned that could be the power cuts, which may affect whole reagent stored in -20°C deep freezer. Is there any alternative strategy for such situation? Addition of solar power backup for the related equipment may be an option to power cuts. Is it possible to share the data with central lab for interpret qPCR data as that can resolve the need of skilled personnel?

Reviewer #2: Author describe the importance of molecular diagnosis using qPCR in a suitcase for rapid malaria surveillance in Ethiopia. The study is designed and conducted well with adequate sample size. However few question need to be answered before considering for publications.

1. the detail protocol of qPCR is missing.

2. author has provided the running cost but no mentioned about the equipment's/ suitcase cost.

3. Author need to mentioned the level of technical expertise required to perform such test.

4. six individual were positive for P. falciparum by microscopy but fail to detect by qPCR, please justify

5. similarly 4 cases were positive for P. vivax by microscopy but fail to detect by qPCR, please justify

6. parasite density need to be mentioned at-least these cases which were missed by qPCR

7. Author may provide the AUC for better understanding the test performance.

8. English language need to be taken care as line no. 241 -242 (Molecular screening for has identified a gap in the sensitivity for diagnosis of clinical P. falciparum cases and a substantial asymptomatic P. vivax reservoir) there is no meaning.

9. Author need to provide the possible couse of fever as only 5% clinical cases were found positive for malaria parasite.

10. Author concluded that Prevention of onward transmission from the asymptomatic P. vivax reservoir might be achieved through improved vector control, however the current manuscript doesn't have objective to asses the vector control measures. Therefore, it can be discussed in the discussion section not to be in conclusion section.

6. PLOS authors have the option to publish the peer review history of their article (what does this mean?). If published, this will include your full peer review and any attached files.

**Do you want your identity to be public for this peer review?** For information about this choice, including consent withdrawal, please see our Privacy Policy.

Reviewer #1: **Yes: **PRASHANT MALLICK

Reviewer #2: No

---

## [Decision Letter · Decision Letter 1]

21 Jun 2022

PGPH-D-22-00611R1

qPCR in a suitcase for rapid Plasmodium falciparum and Plasmodium vivax surveillance in Ethiopia

Dear Dr. Koepfli,

Thank you for submitting your manuscript to PLOS Global Public Health. After careful consideration, we feel that it has merit but does not fully meet PLOS Global Public Health’s publication criteria as it currently stands. Therefore, we invite you to submit a revised version of the manuscript that addresses the points raised during the review process.

Please submit your revised manuscript by . If you will need more time than this to complete your revisions, please reply to this message or contact the journal office at globalpubhealth@plos.org. Please include the following items when submitting your revised manuscript:

We look forward to receiving your revised manuscript.

Kind regards,

Abhinav Sinha, M.D.

Academic Editor

Journal Requirements:

1. We have noticed that you have a list of Supporting Information legends (Supplementary File S1: Laboratory protocols) in your manuscript. However, there are no corresponding files uploaded to the submission. Please upload them as separate files with the item type 'Supporting Information'. 

2. We have noticed that you have uploaded Supporting Information files (Koepfli_PLOSGPH_Questionnaire.docx), but you have not included a list of legends. Please add a full list of legends for your Supporting Information files after the references list. 

Additional Editor Comments (if provided):

Although the authors did a good job in critically revising the manuscript, there are some very minor revision/concerns that are needed to be addressed.

Reviewers' comments:

Reviewer's Responses to Questions

**Comments to the Author**

1. If the authors have adequately addressed your comments raised in a previous round of review and you feel that this manuscript is now acceptable for publication, you may indicate that here to bypass the “Comments to the Author” section, enter your conflict of interest statement in the “Confidential to Editor” section, and submit your "Accept" recommendation.

Reviewer #1: All comments have been addressed

Reviewer #2: All comments have been addressed

2. Does this manuscript meet PLOS Global Public Health’s publication criteria? Is the manuscript technically sound, and do the data support the conclusions? The manuscript must describe methodologically and ethically rigorous research with conclusions that are appropriately drawn based on the data presented.

Reviewer #1: Yes

Reviewer #2: Partly

3. Has the statistical analysis been performed appropriately and rigorously?

Reviewer #1: Yes

Reviewer #2: I don't know

4. Have the authors made all data underlying the findings in their manuscript fully available (please refer to the Data Availability Statement at the start of the manuscript PDF file)?

Reviewer #1: Yes

Reviewer #2: Yes

5. Is the manuscript presented in an intelligible fashion and written in standard English?

Reviewer #1: Yes

Reviewer #2: Yes

6. Review Comments to the Author

Reviewer #1: Reviewer's comments are well-addressed by author.

Reviewer #2: 1. Author advocating the qPCR technology and still putting the methods as supplementary file, I believe it should be in main file.

2.It is evaluation of the techniques so the parasite density must be required which is not done during the study, therefore, I would recommend that this study may consider as pilot study and validation study with comprehensive methodology covering all the aspect need to be done in future. According title of the manuscript may be modified.

3.

7. PLOS authors have the option to publish the peer review history of their article (what does this mean?). If published, this will include your full peer review and any attached files.

**Do you want your identity to be public for this peer review?** For information about this choice, including consent withdrawal, please see our Privacy Policy.

Reviewer #1: **Yes: **PRASHANT MALLICK

Reviewer #2: No

---

## [Editor Report · Decision Letter 2]

27 Jun 2022

qPCR in a suitcase for rapid Plasmodium falciparum and Plasmodium vivax surveillance in Ethiopia

PGPH-D-22-00611R2

Dear Mr. Koepfli,

We are pleased to inform you that your manuscript 'qPCR in a suitcase for rapid Plasmodium falciparum and Plasmodium vivax surveillance in Ethiopia' has been provisionally accepted for publication in PLOS Global Public Health.

Best regards,

Abhinav Sinha, M.D.

Academic Editor